# Enhancing Oxidative Stability and Nutritional Quality of Flaxseed Oil Using Apricot, Sesame, and Black Cumin Oil Blends

**DOI:** 10.3390/foods14112000

**Published:** 2025-06-05

**Authors:** Dino Muhović, Gorica Cvijanović, Marija Bajagić, Lato Pezo, Lazar Pejić, Biljana Rabrenović

**Affiliations:** 1Faculty of Biofarming, Megatrend University, 11070 Belgrade, Serbia; cvijagor@yahoo.com (G.C.); bajagicmarija@yahoo.com (M.B.); 2Institute of General and Physical Chemistry, Studentski Trg 12-16, 11158 Belgrade, Serbia; 3Faculty of Agriculture, University of Belgrade, Nemanjina 6, 11080 Belgrade, Serbia

**Keywords:** oil stabilization, phenolics, flaxseed oil, linseed oil, sesame oil, apricot oil, black cumin oil, oil blend, oxidative stability, antioxidant capacity

## Abstract

There is an unmet need for an affordable, high-quality, and non-thermally processed source of omega-3 fatty acids. Cold-pressed flaxseed oil comes closest to meeting these criteria. Flaxseed oil is also subject to rapid oxidative degradation. Sesame, black cumin, and apricot kernel oils are already used as functional foods and are more resistant to oxidative degradation. GC, HPLC, DPPH, the Folin−Ciocalteu method, and OXITEST were applied to the four cold-pressed oils and their binary blends with flaxseed oil. The fatty acid profile showed that the dominant fatty acid in flaxseed oil was linolenic acid with a content of 52.27 ± 0.17%, while oleic acid dominated in apricot kernel oil (69.45 ± 0.18%) and linoleic acid (58.80 ± 0.07%) in black cumin oil, while sesame oil was characterized by approximately equal proportions of oleic (42.21 ± 0.20%) and linoleic acids (43.37 ± 0.07%). The content of oleic acid showed a moderate, statistically significant correlation with the oxidative stability of oils and blends. The antioxidant capacity of flaxseed oil (25 ± 1.4 μmol TE/g) was most strongly influenced by the addition of black cumin oil (75 ± 3.5 μmol TE/g), so that the highest antioxidant capacity was achieved by the blend with an addition of 50% of this oil (57.5 ± 2.4 μmol TE/g). Oxidative stability tests show that apricot kernel oil stabilizes flaxseed oil the most and increases the oxidative stability of the blend by up to 60%.

## 1. Introduction

The presence of polyunsaturated fatty acids (PUFAs) in vegetable oils, especially omega-3 fatty acids, is highly desirable, as they contribute to nutritional value and have certain health benefits. However, polyunsaturated fatty acids, such as alpha-linolenic acid, have a methylene group (-CH_2_-) that lies between two double bonds (a double allylic position) and is highly reactive. Under conditions where oxygen is present, alpha-linolenic acid reacts in a process called auto-oxidation, forming a lipid radical [1]. Such oxidation leads to a chain reaction of free radicals that initiate the process of lipid peroxidation [1], in which reactive carbonyl compounds such as malondialdehyde are formed in addition to lipid peroxides [2]. The lipid peroxidation products can cause damage to a living organism by damaging biomolecules through reactive carbonyl species. In this process, a carbonyl group reacts with an amino group in a protein or nucleic acid. One of the consequences of this reaction is DNA damage, which can lead to mutagenesis and, in some cases, cancer [3].

Flaxseed oil is a vegetable oil obtained from flax seed (*Linum usitatissimum* L.). This oil is one of the most important sources of alpha-linolenic acid [1,2,3,4,5,6,7,8]. In addition to omega-3 fatty acid, which is alpha-linolenic, flaxseed oil is also rich in omega-6 fatty acid, which is linoleic, albeit to a lesser extent [4,5,6,7,8,9,10,11]. This highly unsaturated oil is also rich in tocopherols, which are the most important protective components of this oil against oxidation. Tocopherols scavenge free radicals and become free radicals themselves, but are stabilized by a resonance structure. Besides tocopherols, natural antioxidants in flaxseed include lignans, which are poorly fat-soluble, which has led to documented efforts to modify them to make them fat-soluble [12]. In addition to their antioxidant effects, lignans in flaxseed were also described in the early medical and biomedical literature as helpful in conditions associated with metabolic syndrome [13,14,15,16]. Numerous studies have shown that flaxseed oil has a positive effect on cardiovascular health due to its high alpha-linolenic acid content, influencing blood lipid levels, blood pressure, endothelial function, and inflammation [17,18,19]. One study reported that the daily consumption of two tablespoons of flaxseed oil over a period of twelve weeks significantly reduced total cholesterol levels in men [20]. In the early clinical literature, it was found to improve verbal fluency in the elderly [21].

The high content of polyunsaturated fatty acids and the subsequent susceptibility to oxidation are factors that limit the use of flaxseed oil in the diet and in the food industry, and also affect the storage conditions and shelf life of the product [22].

Considerable efforts were made in the scientific literature to stabilize omega-3 fatty acid sources through encapsulation [23] and various synthetic and natural antioxidants [7,24,25]. The addition of artificial and sometimes even natural antioxidants may be undesirable. One alternative to additives is blending vegetable oils. The blending of oils affects the triacylglycerol profile, and thus indirectly the oxidative stability, and can also lead to changes in the content of bioactive components. Romanić et al. [26] improved the ratio of omega-6 and omega-3 fatty acids by adding refined sunflower oil to cold-pressed flaxseed oil, and the color and sensory characteristics of the resulting blend were satisfactory.

Apricot kernel oil contains more than 80% monounsaturated fatty acids, including 60–70% oleic acid, as reported by Stryjecka et al. [27]. Due to its favorable fatty acid composition and the presence of γ-tocopherol as the dominant phenolic compound, this oil has very good oxidative stability and shelf life, with an induction time of 19.7 h as determined by the Rancimat test at 110 °C [28].

Sesame seed oil contains a high proportion of unsaturated fatty acids, in particular oleic and linoleic acid, and tocopherols and tocotrienols, and is also characterized by two lignans, sesamin and sesaminol, which not only contribute to the oxidative stability and good thermal behavior [29] of this oil, but also have significant health effects, such as a reduction in markers of metabolic syndrome [30]. According to Gharby et al. [31], the induction period for sesame seed oil investigated using the Rancimat method was 28.51 h at 110 °C.

Black cumin oil is rich in polyunsaturated fatty acids, the most important of which is linoleic acid (C18:2) [32]. It contains a secondary metabolite, thymoquinone, which causes most of its biological activity, as well as its excellent oxidative stability and antioxidant activity [33,34]. Thymoquinone has already shown a protective effect against oil oxidation [35], making black cumin oil a reasonable choice for blending with flaxseed oil.

Blending oils can be a simple and acceptable method to improve nutritional composition and to obtain certain physicochemical properties of the resulting blends. The aim of this study was to formulate binary blends of cold-pressed flaxseed oil (in the form of 70 and 50 wt% flaxseed oil) with cold-pressed apricot kernel oil, sesame oil, and black cumin oil to achieve a balanced fatty acid profile, good oxidative stability, and a high content of bioactive compounds. This research is unique, as such blends have not been studied before and can serve as a starting point for further research and potential commercialization.

## 2. Materials and Methods

To ensure the authenticity of the oil, apricot kernels, flaxseed, and sesame seeds were purchased from grain wholesalers (Belgrade, Serbia). Unfortunately, there were no good-quality black cumin seeds available, so cold-pressed oil from Turkey was used. All standardized chemicals used for this study were of a p.a. grade, and thus they were not additionally purified. They were purchased from Merck Science Life S.R.L. (Belgrade, Serbia).

### 2.1. Cold Pressing

The extraction of oils from the seed samples was performed using a Kern Kraft KK40F screw press (Benenv Co., Ltd., Baden-Württemberg, Germany). During pressing, the highest temperature of the outlet oil was below 45 °C, which corresponds with the recommendation to perform cold pressing at a temperature below 50 °C in order to preserve the bioactive components [36]. After 24 h of settling, the oils were filtered through filter paper, transferred to glass bottles under the inert gas atmosphere, and stored in the refrigerator for further analysis.

### 2.2. Oil Blending

A mixture of 100 mL was prepared by mixing flaxseed oil (F) with each cold-pressed oil (sesame oil—S, apricot kernel oil—A, and black cumin oil—BC) in two ratios (70:30 and 50:50 m/m) using a mechanical stirrer at 180 rpm for 5 min [37].

### 2.3. Determination of the Fatty Acid Composition

The fatty acid composition of the oils and blends was determined by gas chromatography according to the standard method [38] on a GC 6890 instrument (Agilent Technologies, Santa Clara, CA, USA) with a split-splitless injector and a flame ionization detector (FID).

The fatty acid methyl esters were prepared according to the standard method [39], and a Supelco SP-2560 capillary column with the following dimensions was used for their separation: length 100 m × inner diameter 0.25 mm × film thickness 0.20 µm (Supelco, Bellefonte, PA, USA), with a flow rate of 5 mL/min, and using a helium phase. The injector temperature was 250 °C, and the detector temperature was 260 °C. The injection volume was 1 µL, and the distribution ratio of the injector was set to 20:1. The column temperature was programmed from an initial 50 °C (for 5 min) to 240 °C (for 20 min), with a linear temperature change of 4 °C/min. The chromatographic peaks in the samples were identified by comparing the relative retention times of the fatty acid methyl esters of samples with the Supelco 37 Component FAME Mix, a mixture of methyl esters (Supelco, Bellefonte, PA, USA). The results were presented as a mean value ± standard deviation (*n* = 2). Further details are provided in the Appendix A.

### 2.4. Determination of Tocopherol Composition

Gimeno et al.’s [40] method was used to determine the tocopherol content in oils. HPLC, normal phase (Agilent 1260 Infinity, Santa Clara, CA, USA), was used in order to determine the amount of individual tocopherol isomers. XBridge C18 (4.6 × 150 mm, 3.5 µm particle size) was the column used. The analytical separation of the tocopherol isomers was achieved with an isocratic elution of methanol–water (96:4, *v*/*v*). The injection volume was 20 µL. The total run time was 5 min, and 2.0 mL/min was the flow rate. Detection was performed with a DAD detector. Tocopherols were monitored at 292 nm, and every run lasted 6 min. Hexane (1:10) was the solvent used for oil dilution. Subsequently, 200 µL was poured into a screw-cap tube, and then 800 µL of methyl alcohol was added. After vortex mixing and centrifugation (3000× *g*, 5 min), a 0.22 µm syringe filter was used to filter the sample. An external calibration curve was generated for each tocopherol standard to calculate the concentration of individual tocopherols existent in the oil sample [41]. The results are expressed as the mean ± standard deviation of two replicates. Further details can be found in the Appendix A.

### 2.5. Oxidative Stability—The OXITEST Method

The oxidative stability of the oil samples was measured with the OXITEST apparatus (Velp Scientifica, Usmate Velate MB, Italy). An oil sample was filled into a hermetically sealed titanium chamber with a thermostat. The OXITEST reactor subjected the oil samples to an accelerated oxidation process by heating them to 90 °C and exposing them to an oxygen pressure of 600 kPa. The instrument was controlled using original software (OXISoft^TM^ 5.0.6, Velp), which monitored the oxygen pressure. At the end of the test, the program automatically calculated the induction period (IP) from the resulting oxidation curves using the two-tangent method, i.e., the time required to reach the starting point of oxidation, which corresponds to a sudden change in the oxygen consumption rate.

### 2.6. Thrombogenicity and Atherogenicity Index

Indexes of thrombogenicity and atherogenicity are indicators of the nutritional quality of oils, which are calculated on the basis of the fatty acid composition. The thrombogenicity index (TI) and the atherogenicity index (AI) are calculated according to the following equations [42].$$TI=\frac{C14:0+C16:0+C18:0}{\left(0.5\cdot \sum MUFA\right)+\left(3\cdot \sum \omega -3\right)+\left(0.5\cdot \sum \omega -6\right)+(\sum \omega -3/\sum \omega -6)}$$$$AI=\frac{C12:0+4\cdot C14:0+C16:0}{\sum MUFA+\sum \omega –3+\sum \omega -6}\;$$
where C14:0 is myristic acid (MA0) as a fraction of the total mass expressed as a percentage, C16:0 is palmitic acid (PA0) as a fraction of the total mass expressed as a percentage, C18:0 is stearic acid (SA0) as a fraction of the total mass expressed as a percentage, MUFA is monounsaturated fatty acids as a fraction of the total mass expressed as a percentage, ω-3 is omega-3 fatty acids as a fraction of the total mass expressed as a percentage, ω-6 is omega-6 fatty acids as a fraction of the total mass expressed as a percentage, and C12:0 is lauric acid as a fraction of the total mass expressed as a percentage.

### 2.7. Measurement of the Antioxidant Capacity

A DPPH assay was used to measure the antioxidant scavenging capacity towards the DPPH radical as described by Chew et al. [43]. The modifications proposed by Kua et al. were also used [44]. Another modification was the sample used, as different samples were tested. In total, 0.1 mL of oil or a blend thereof was added to 3.9 mL of ethanolic DPPH (2.4 mg/100 mL ethanol). The mixture was shaken for 1 min and incubated for 30 min at room temperature in the dark. The resulting solution was measured at 517 nm with a spectrophotometer (Thermo Scientific Evolution 600 UV-Vis, Waltham, MA, USA) against a blank solution (absolute ethanol). The results are expressed as μmol of Trolox equivalents per gram of sample (μmol TE/g). The antioxidant capacity was determined in five replicates. Detailed information is given in the Appendix A.

### 2.8. Total Phenolic Compounds

The Folin–Ciocalteu spectrophotometric method, following the method of Radočaj et al. [45], which in turn is a synthesis of the protocols of Haiyan et al. [46], as well as Rabrenović et al. [36], was used to measure the content of total phenols. In brief, the oil was dissolved in hexane and then extracted with methanol. The sample was left to stand overnight. The methanolic extract was then washed with hexane, and an aliquot of 1 mL was transferred to a 10 mL volumetric flask and mixed with the Folin–Ciocalteu reagent. After shaking, it was allowed to stand for 3 min, and then an aqueous saturated sodium carbonate solution was added, and then diluted with water. One hour later, the absorbance was measured at a wavelength of 725 nm compared to a blank using a UV/VIS spectrophotometer model T80þ (PG Instruments Limited, London, UK). Gallic acid was used for calibration. The results were expressed as mg of gallic acid equivalent per gram of sample (mg GAE/g).

### 2.9. Principal Phenolic Compound Measurement

For sample preparation, the LLE1 protocol was used as established by Tasioula et al. [47]. For the analysis of phenolic compounds, the protocol described by Mizzi et al. was used [48]. Reversed-phase high-performance liquid chromatography (HPLC) with UV/VIS detection was performed using a Waters 2695 Alliance HPLC system (Waters Inc., Milford, CT, USA). The phenolic compounds were separated using a C18 column (Waters SunfireTM) with the dimensions 150 × 4.6 mm and a particle size of 5 µm. Acetonitrile (phase A) and 0.1% phosphoric acid solution in deionized water (phase B) were used as mobile phases. The samples were analyzed using the gradient of the mobile phase with the following steps as previously explained [48]: “(a) initial 5% A and 95% B, (b) 15 min 35% A and 65% B, (c) 20 min 35% A and 65% B, (d) 30 min 40% A and 60% B, (e) 35 min 40% A and 60% B, (f) 40 min 50% A and 50% B, (g) 52 min 70% A and 30% B, and (h) 60 min 5% A and 95% B. A constant flow rate of 0.5 mL/min and a temperature of 5 °C were used”. The main phenolic compounds were detected at the specific wavelengths for each compound. Secoisolariciresinol diglucoside from flaxseed oil and sesamin and sesamolin from sesame oil were measured at 280 nm. In the case of apricot kernel oil, flavonoids were detected at 320 nm and hydroxycinnamic acid at 350 nm. Thymoquinone and carvacrol from black cumin seed oil were detected at 254 nm and 220 nm, respectively.

### 2.10. Statistical Analysis and Description of the Results

Data distribution normality was evaluated using the Shapiro–Wilk and Anderson–Darling tests. The results are expressed as mean values accompanied by standard deviations. Tukey’s HSD test was applied to assess differences among sample means. The test was used to determine statistically significant differences between means of different fatty acid residues. All of the data were subjected to thorough statistical analysis, including descriptive statistics and Pearson’s correlation analysis, using the STATISTICA 10.0 software package (StatSoft Inc., Tulsa, OK, USA). Pearson correlation coefficients with significance levels were used to assess the relationships between fatty acid residues. Pearson correlation was also used to test the correlation between phenolics and antioxidant capacity and the correlation between oleic acid and the induction period. Cluster analysis (CA) and Principal Component Analysis (PCA) were employed as multivariate statistical methods for data exploration and pattern identification. CA classifies objects based on their similarities, facilitating the detection of underlying structures within complex datasets. PCA, on the other hand, reduces data dimensionality by converting correlated variables into a smaller number of uncorrelated principal components, thereby retaining the most significant variance and improving data interpretability.

### 2.11. Use and Potential Use of AI

Due to the current insufficient reliability of generative AI, no single sentence in this manuscript was written by it. AI technology was used or potentially used in the following ways. Scribbr Free Citation Generator was used for reference management. It failed to abbreviate journal names and failed to differentiate between different types of sources. All sources were wrongly cited as journals instead. The tool is unlikely to be based on generative AI, but such a possibility could not have been reliably excluded. For spell-checking and grammar editing, as well as converting accidental British spelling into American spelling, QuillBot was used. The writing score on QuillBot is probably based on AI, but if so, it is not generative. It should be noted that while QuillBot contains a generative AI-based feature to paraphrase authors’ sentences, that feature has not been used. Claude 3.7 Sonnet was prompted to replace the word “signature” in “signature phenolic compounds” with another word of a similar meaning. It created a list of words, and the word “principal” from that list was used, so the Methods section was called “Principal Phenolic Compounds”. Claude 3.7 Sonnet and ChatGPT4o were used for preliminary reference management. The ISO 4 abbreviation was preliminarily generated using these two tools. The CAS Source Index (CASSI) Search Tool link was generated by ChatGPT4o. Lastly, Claude 3.7 Sonnet was used to detect sentences in the present perfect tense, as this tense is discouraged by the rules of the publisher but not detected by QuillBot, as the sentences are grammatically correct. It was tasked to copy sentences with present perfect in them in two separate threads. It found six actual cases of present perfect tense and produced two false positives. The second time, it produced eight elements on a list, and all were false positives. It failed to spot two cases of present perfect tense use. The sentences found by Claude 3.7 Sonnet and the sentences it failed to find were rewritten without GenAI, as the AI was prompted merely to copy the sentences rather than modify them.

## 3. Results and Discussion

### 3.1. Fatty Acid Composition and Nutritive Indices

From the results in Table 1, it can be concluded that the predominant fatty acid in flaxseed oil (F oil) was linolenic acid (LA3) with a content of 52.27 ± 0.17%, and of the polyunsaturated fatty acids, linoleic acid (LA2) was also present with a content of 14.33 ± 0.08%. The oleic acid (OA1) was found in the sample with a content of 22.91 ± 0.11%. Of the saturated fatty acids, palmitic acid (PA0) was predominant with a content of 5.82 ± 0.05%. The results obtained are in full agreement with the results in the literature [11,49,50].

Polyunsaturated fatty acids dominate in this vegetable oil, which was also the case with BC oil. However, linoleic acid, an omega-6 fatty acid, dominates in BC oil with a content of 58.80 ± 0.07%. The monounsaturated oleic acid was just behind linoleic acid with a content of 24.93 ± 0.11%, and this oil is characterized by a high proportion of saturated palmitic acid, which is 13.11 ± 0.05%. All these findings are consistent with the results reported in the literature [31,32].

Apricot kernel oil (A oil) was also analyzed, in which, as can be seen from Table 1, monounsaturated fatty acids were dominant, primarily oleic acid with a content of 69.45 ± 0.18%, then with significantly lower proportions of palmitoleic acid and heptadecanoic acid. This makes A oil very suitable for blending, as it increases the share of monounsaturated acids. In addition to oleic acid, A oil also contained polyunsaturated linoleic acid with a content of 23.57 ± 0.09%. Makrygiannis et al. [51] reported similar results for apricot kernel oil, 65.71% for oleic acid, and 28.16% for linoleic acid.

The sesame oil (S oil) had a specific fatty acid composition, as it is an oil of the oleic-linoleic type, in which these two acids were present in almost equal proportions, as is the case with pumpkin oil or corn germ oil [52,53]. This was exactly the case with the S oil examined. The oleic acid content was 42.21 ± 0.20%, while the linoleic acid (LA2) content was 43.37 ± 0.07%.

In order to improve the oxidative stability of the F oil and increase its nutritional value, it was mixed with each of the cold-pressed oils mentioned in a ratio of 70:30 and 50:50.

This naturally led to changes in the fatty acid composition of the resulting blends, especially in blends with equal proportions (50:50). The greatest changes occurred in the contents of oleic acid, linoleic acid, and linolenic acid. By increasing the share of A oil in the blends, the content of oleic acid was significantly increased (*p* ˂ 0.05). The highest level of oleic acid was found in the flaxseed–apricot kernel oil blend 50% (FA50) (45.96 ± 0.17%). When the proportion of S oil and BC oil was increased in blends, the content of omega-6 linoleic acid rose statistically significantly (*p* ˂ 0.05). The FBC50 blend had the highest content of linoleic acid, 35.72 ± 0.08%. This was also reflected in the total content of saturated fatty acids (SFAs), monounsaturated fatty acids (MUFAs), and polyunsaturated fatty acids (PUFA).

The sum of ΣMUFA had the highest value in A oil (70.46 ± 0.16%), while FA50 and FA73 had 46.57 ± 0.18% and 36.81 ± 0.13%, respectively. ΣPUFA was highest in F oil (66.60 ± 0.08%), followed by FBC73 and FBC50 with 64.76 ± 0.07% and 63.18 ± 0.03%, respectively.

The PUFA/SFA ratio is often used as an indicator of the atherogenicity of a particular fat or oil. However, some studies document that not all fatty acids have the same effect on LDL and HDL levels. Among the long-chain SFAs, myristic acid stands out as the fatty acid with the most extreme atherogenic effect, which is four times higher than that of lauric acid and palmitic acid, while stearic acid and saturated fatty acids with ten or fewer carbon atoms have no effect [54]. Therefore, Ulbricht and Southgate [42] proposed including in the PUFA/SFA ratio only the pro-atherogenic SFAs, which are considered to be the cause of coronary disease, but also to include the MUFAs in the calculation, thus obtaining the atherogenic index. The same authors proposed another nutritional indicator—the thrombogenicity index (TI), in the calculation of which all thrombogenic SFAs (myristic, palmitic, and stearic acids), MUFAs, and PUFAs that show an antithrombogenic effect, as well as the ratio of PUFAs from the n-3 and n-6 series, are included. Different coefficients indicate that MUFAs and PUFAs from the n-6 series do not have a strong antithrombogenic effect, as is the case with PUFAs from the n-3 series. Lower values of AI and TI indicate better nutritional properties and vice versa. The AI of the investigated oils and their blends ranged from 0.06 to 0.19 (Table 1). Apricot oil had the lowest AI value, while black cumin oil had the highest. Among the oil blends with the lowest AI value, the FA50 blend stood out with an AI of 0.08. The analyzed samples also had optimal TI values, ranging from 0.06 to 0.32. The TI of the blends ranged from 0.06 to 0.11. For comparison, Benkhoud et al. [55] reported values of 0.16 and 0.30 for AI and TI for extra virgin olive oil (EDMU), respectively.

### 3.2. Tocopherol Content

Due to their high solubility in fats and oils, tocopherols are considered to be the most important antioxidants responsible for the stability of vegetable oils. Although other antioxidants can also play an important role, the oxidative stability of the oil frequently depends on the presence of tocopherols, particularly the presence of α tocopherol (ALPHA) and γ tocopherol (GAMMA). For ALPHA, A oil displayed the maximum value (2.38 ± 0.06 mg/100 g), with FA50 and FA73 showing 1.70 ± 0.04 mg/100 g and 1.43 ± 0.03 mg/100 g, respectively. β tocopherol (BETA) reached its maximum in A oil (3.08 ± 0.06 mg/100 g). GAMMA was dominant in all single oils, as well as in the blends, with the highest value in S oil (38.98 ± 0.18 mg/100 g), followed by A oil (37.36 ± 0.11 mg/100 g), and the FS50 oil blend (34.16 ± 0.13 mg/100 g). The results obtained are in complete agreement with the values given in the literature for F, A, S, and BC oils [32,56,57,58].

### 3.3. Correlation Analysis

The correlation matrix revealed several statistically significant relationships (*p* < 0.05) among the variables (Figure 1). A strong positive correlation was observed between palmitic acid and linoleic acid (*r* = 0.968, *p* < 0.001), as well as between heptadecanoic acid and PUFAs (*r* = 0.963, *p* < 0.001). Additionally, MUFAs and PUFAs exhibited a strong negative correlation (*r* = −0.986, *p* < 0.001), while gondoic acid (GA1) was strongly negatively correlated with SFAs (*r* = −0.939, *p* < 0.001). Moderate positive correlations included MA0 with PA0 (*r* = 0.739, *p* = 0.015) and gondoic acid (*r* = 0.974, *p* < 0.001), as well as heptadecanoic acid with MUFAs (*r* = 0.835, *p* = 0.003). Significant negative correlations included palmitoleic acid with oleic acid (*r* = −0.849, *p* = 0.002) and gondoic acid (*r* = −0.862, *p* = 0.001), and stearic acid with linoleic acid (*r* = −0.798, *p* = 0.006) and PUFA (*r* = −0.886, *p* = 0.001).

### 3.4. Cluster Analysis

This clustering structure suggests that the sample in each cluster may be affected by different variables, while the separation of F points to its independent or anomalous nature within the dataset.

The results of the cluster analysis indicate distinct groupings of the studied variables (myristic acid, palmitic acid, palmitoleic acid, heptadecanoic acid, heptadecenoic acid (HE1), stearic acid, oleic acid, linoleic acid, linolenic acid, arachidic acid (AA0), gondoic acid, ΣSFA, ΣMUFA, ΣPUFA, ALPHA, BETA, GAMMA) based on their similarities (Figure 2). The first cluster, comprising samples FBC50, FBC73, and FS50, showed that these samples share similar patterns due to a similar content of stearic acid (SA0), ΣSFA, and ΣPUFA. The second cluster, which includes 70% flaxseed, 30% sesame oil blend (FS73), FA50, FA73, and BC, indicates a closely related content of GAMMA.

The third cluster, consisting of S and A, implies that these two samples exhibit similar content of oleic acid, heptadecanoic acid, heptadecenoic acid, ΣMUFA, palmitoleic acid, ALPHA, and BETA content. Lastly, F, being isolated, indicates it is significantly different from all other samples, possibly due to the unique content of linolenic acid and stearic acid.

### 3.5. Principal Component Analysis

The results of the Principal Component Analysis (PCA) reveal the following numerical influences based on the correlation structure. PC1 shows that myristic acid (MA0) (3.78%), palmitic acid (5.52%), SFAs (8.79%), and PUFAs (7.44%) have positive loadings, suggesting their positive influence along this axis (Figure 3). In contrast, palmitoleic acid (−9.73%), heptadecanoic acid (−9.75%), heptadecenoic acid (−4.65%), and ALPHA (−6.24%) have negative loadings, indicating an inverse relationship along PC1. PC2 highlights that stearic acid (11.75%), linolenic acid (11.92%), and GAMMA (10.26%) have higher positive scores, indicating a strong influence in the positive direction of this component. Conversely, myristic acid (−13.84%), gondoic acid (−14.02%), palmitic acid (−8.10%), and palmitoleic acid (−1.38%) show negative scores along PC2, implying a contrasting relationship with these variables along this axis. The rest of the variables, such as SFAs (1.14%), MUFAs (−0.79%), PUFAs (1.59%), BETA (−1.85%), and ALPHA (−4.56%), have intermediate values, suggesting moderate influences along both components. These results indicate that PC1 captures a significant contrast between certain variables, while PC2 reveals a distinct grouping of variables with positive or negative associations.

### 3.6. Total Phenolic Content and Antioxidant Capacity

The content of phenolic compounds and antioxidant capacity of the oils were determined (Table 2). BC oil showed the highest phenolic content (2.5 mg GAE/g) and antioxidant capacity (75 µmol TE/g), followed by S oil (1.2 mg GAE/g and 30 µmol TE/g), F oil (1 mg GAE/g and 25 µmol TE/g), and A oil (0.8 mg GAE/g and 20 µmol TE/g) in decreasing amounts for both properties. The binary blends of flaxseed oil with each of the other three oils were also analyzed (Table 2). FBC50 was found to exhibit the highest phenolic content (1.75 mg GAE/g) and antioxidant capacity (57.5 µmol TE/g) of all the blends. The flaxseed–apricot kernel oil blends had the lowest values for both antioxidant capacity and phenolic content: 24.5 µmol TE/g and 0.9 mg GAE/g for the FA50 and 24.7 µmol TE/g, and 0.9 mg GAE/g for the FA73. Pearson correlation analysis of total phenols and antioxidant capacity yielded an R value of 0.989 and *p* < 0.00001.

### 3.7. Principal Phenolic Compounds

Principal phenolic compounds were detected in each single oil (Table 3). Sesame oil was found to contain its principal metabolites, sesamin (4.10 mg/g) and sesamolin (0.85 mg/g). Black cumin oil was found to contain thymoquinone (2.1 mg/g) and small amounts (0.2 mg/g) of carvacrol. The flaxseed oil sample contained secoisolariciresinol diglucoside at 0.29 mg/g oil. Johnsson et al. [59] reported the content of this lignan in flaxseed can range from 610 to 1300 mg/100 g. This is an unusually low amount of thymoquinone, but it has been scientifically documented several times that thymoquinone levels can be unusually low [60]. Apricot kernel oil was found to contain hydroxycinnamic acids (0.20 mg/g oil) and flavonoids (0.17 mg/g oil).

### 3.8. Oxidative Stability

Oxidative stability is defined as the resistance of oils and fats to oxidative changes during production and storage [61]. Although it is not a standard parameter for oil quality, oxidative stability is a good indicator of shelf life that is useful for evaluating oil quality. The composition of the FAs, i.e., the share of those with more unsaturated bonds, largely determines the oxidative stability of the oil. The higher their proportion, the lower the oil’s resistance to oxidation. However, the antioxidative compounds present, such as ALPHA and GAMMA tocopherol, also play an important role in the oxidative stability of the oil. The oxidative stability of the oil samples, expressed as the induction period (IP) in hours (h), was tested with an Oxitest device.

As shown in Table 1 and Figure 1, A oil exhibited the longest induction period (55.17 h), followed by S oil (33.47 h), and BC oil (30.49 h), and F oil had the shortest induction period (7.15 h). The literature contains numerous data on the IPs for these oil types, which were determined using the Rancimat test, but due to the different methodology, a direct comparison with the results of the Oxitest was not possible. Symoniuk et al. [52] give the IP determined with the Rancimat test for flaxseed oil as 3.84–4.65 h at temperatures of 100 °C and for black cumin oil as 13.45–38.42 h. For example, Edris [62] reported an IP of crude black cumin oils between 2.5 h and 26.9 h obtained by the Rancimat test at 100 °C and an air flow rate of 20 mL/min. Sahrae Ardakani et al. [63] tested sesame oil using the Rancimat test at 120 °C and an air flow rate of 20 mL/min, and found that the IP was 11.02 h. According to Uluata [64], cold-pressed apricot oil had an IP of 15.1 h under the Rancimat test conditions of 110 °C and an air flow rate of 20 mL/min.

Since the aim was to improve the oxidative stability of F oil, this goal was achieved by adding different proportions of S, A, and BC oil. The best oxidative stability was achieved in the FA50 oil blend, and oleic acid showed a moderately high Pearson correlation with an induction period of 0.7577 (*p* = 0.011). The IP of this blend was 11.42 h, which meant a 60% better oxidative stability compared to the initial F oil. Although the FS50 and FBC50 blends had a high content of linoleic acid, whose oxidation rate is 22 times higher than that of oleic acid [65], they showed a very good oxidative stability of 10.53 and 10.35 h, respectively, which can be explained by the presence of the individual antioxidant components such as the sesame oil lignans, sesamin and sesamolin, and the thymoquinone in the black cumin oil, whereby the synergistic effect should also be taken into account. Yamashita [66] states that there is a strong synergistic effect between sesamin and sesamolin with gamma-tocopherol, which dominates in sesame oil.

## 4. Conclusions

The results of testing binary blends of flaxseed oil (F) with apricot kernel oil (A), sesame seed oil (S), and black cumin seed oil (BC) with shares of 30% and 50% showed an improvement in most physico-chemical properties compared to flaxseed oil. The blends of sesame oil and black cumin oil with flaxseed oil showed improvements in oxidative stability, total phenolic compounds, and antioxidant capacity, which, in the case of the blend FBC50, the best blend in terms of these two properties, exceeded the antioxidant capacity of flaxseed oil by 2.3-fold. Although the FA73 and FA50 blends had a lower antioxidant capacity compared to the other blends tested, they showed significantly improved oxidative stability, with the FA50 blend achieving a 60% longer induction time compared to F oil. The increased induction period correlated with and was likely caused by oleic acid. The results indicate that oil blending is a viable strategy to stabilize flaxseed oil and improve its antioxidant capacity. Further studies to evaluate the thermal behavior and sensory properties of these oil blends are needed to expand their applicability in various food and nutraceutical formulations.

## Figures and Tables

**Figure 1 foods-14-02000-f001:**
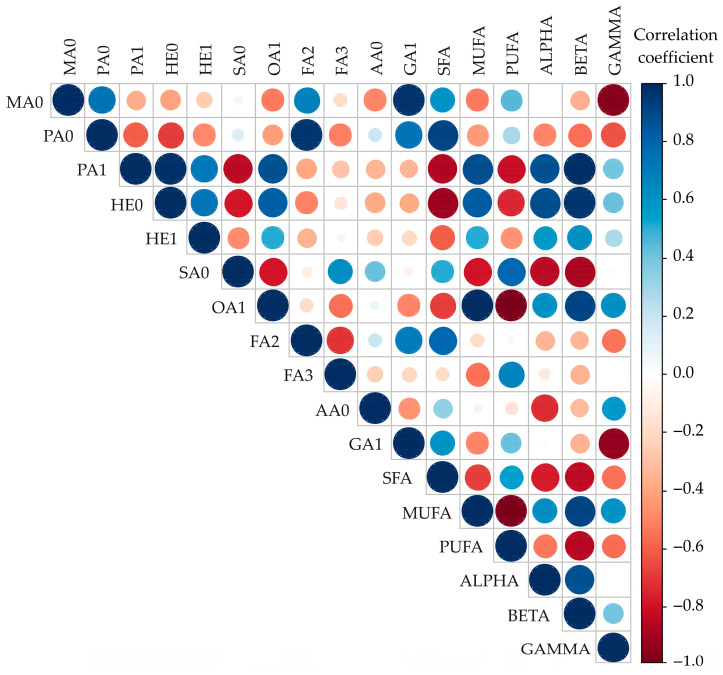
Correlation analysis of chemical compounds obtained for sampling. MA0—myristic acid (C14:0); PA0—palmitic acid (C16:0); PA1—palmitoleic acid (C16:1); HE0—heptadecanoic acid (C17:0); HE1—heptadecenoic acid (C17:1); SA0—stearic acid (C18:0); OA1—oleic acid (C18:1); LA2—linoleic acid (C18:2); LA3—linolenic acid (C18:3); AA0—arachidic acid (C20:0); GA1—gondoic acid (C20:1); ΣSFA—sum of saturated fatty acids; ΣMUFA—sum of monounsaturated fatty acids; ΣPUFA—sum of polyunsaturated fatty acids; ALPHA—α tocopherol; BETA—β tocopherol; GAMMA—γ tocopherol.

**Figure 2 foods-14-02000-f002:**
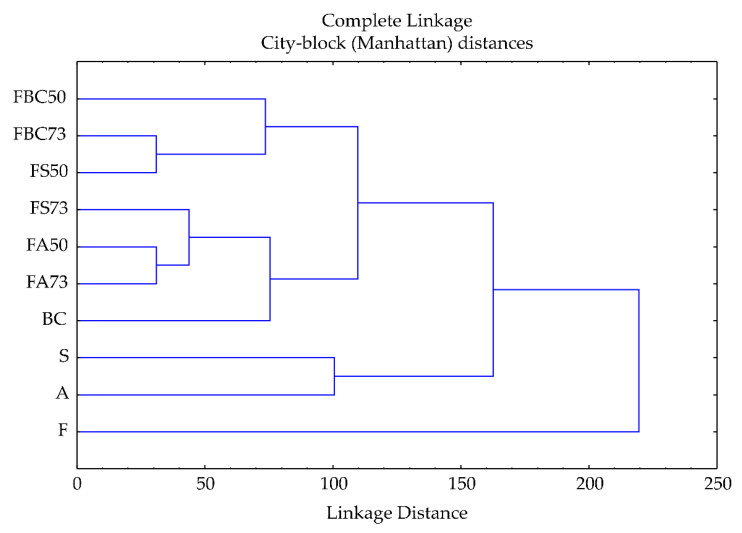
Cluster analysis. F—flaxseed oil; A—apricot kernel oil; S—sesame seed oil; BC—black cumin oil; FA73, FA50, FS73, FS50, FBC73, FBC50—oil blends in proportions 70:30 and 50:50.

**Figure 3 foods-14-02000-f003:**
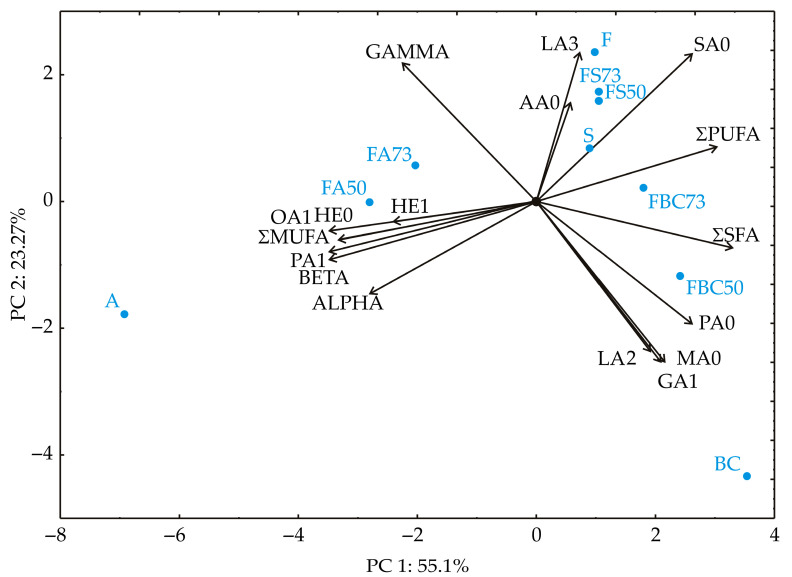
PCA of the samples. F—flaxseed oil; A—apricot kernel oil; S—sesame seed oil; BC—black cumin oil; FA73, FA50, FS73, FS50, FBC73, FBC50—oil blends in proportions 70:30 and 50:50. MA0—myristic acid (C14:0); PA0—palmitic acid (C16:0); PA1—palmitoleic acid (C16:1); HE0—heptadecanoic acid (C17:0); HE1—heptadecenoic acid (C17:1); SA0—stearic acid (C18:0); OA1—oleic acid (C18:1); LA2—linoleic acid (C18:2); LA3—linolenic acid (C18:3); AA0—arachidic acid (C20:0); GA1—gondoic acid (C20:1); ΣSFA—sum of saturated fatty acids; ΣMUFA—sum of monounsaturated fatty acids; ΣPUFA—sum of polyunsaturated fatty acids; ALPHA—α tocopherol; BETA—β tocopherol; GAMMA—γ tocopherol.

**Table 1 foods-14-02000-t001:** Fatty acid composition, nutritional value indices, tocopherol content, and induction period of the analyzed samples.

Component (%)	F	A	S	BC	FA73	FA50	FS73	FS50	FBC73	FBC50
MA0	0.04 ± 0.00 ^c^	0.02 ± 0.00 ^ab^	0.01 ± 0.00 ^a^	0.19 ± 0.01 ^f^	0.04 ± 0.00 ^c^	0.03 ± 0.00 ^bc^	0.04 ± 0.00 ^c^	0.03 ± 0.00 ^bc^	0.08 ± 0.00 ^d^	0.12 ± 0.01 ^e^
PA0	5.82 ± 0.05 ^b^	5.21 ± 0.06 ^a^	10.06 ± 0.08 ^f^	13.11 ± 0.05 ^g^	5.35 ± 0.06 ^a^	5.43 ± 0.09 ^a^	6.88 ± 0.06 ^c^	7.82 ± 0.08 ^d^	7.90 ± 0.11 ^d^	9.27 ± 0.11 ^e^
PA1	0.01 ± 0.00 ^a^	0.93 ± 0.02 ^d^	0.01 ± 0.00 ^a^	0.01 ± 0.00 ^a^	0.49 ± 0.02 ^b^	0.53 ± 0.01 ^c^	0.01 ± 0.00 ^a^	0.01 ± 0.00 ^a^	0.01 ± 0.00 ^a^	0.01 ± 0.00 ^a^
HE0	0.04 ± 0.00 ^a^	0.10 ± 0.01 ^c^	0.03 ± 0.00 ^a^	0.03 ± 0.00 ^a^	0.07 ± 0.01 ^b^	0.07 ± 0.00 ^b^	0.04 ± 0.00 ^a^	0.03 ± 0.00 ^a^	0.04 ± 0.00 ^a^	0.03 ± 0.00 ^a^
HE1	0.01 ± 0.00 ^a^	0.03 ± 0.00 ^b^	0.01 ± 0.00 ^a^	0.01 ± 0.00 ^a^	0.04 ± 0.00 ^c^	0.01 ± 0.00 ^a^	0.01 ± 0.00 ^a^	0.01 ± 0.00 ^a^	0.01 ± 0.00 ^a^	0.01 ± 0.00 ^a^
SA0	4.43 ± 0.03 ^h^	0.54 ± 0.01 ^a^	3.83 ± 0.06 ^f^	2.48 ± 0.02 ^c^	3.21 ± 0.04 ^d^	2.28 ± 0.04 ^b^	4.29 ± 0.04 ^gh^	4.16 ± 0.04 ^g^	3.87 ± 0.04 ^f^	3.44 ± 0.05 ^e^
OA1	22.91 ± 0.11 ^a^	69.45 ± 0.18 ^i^	42.21 ± 0.20 ^g^	24.93 ± 0.11 ^c^	36.20 ± 0.11 ^f^	45.96 ± 0.17 ^h^	28.42 ± 0.10 ^d^	32.14 ± 0.12 ^e^	23.16 ± 0.08 ^a^	23.76 ± 0.08 ^b^
LA2	14.33 ± 0.08 ^a^	23.57 ± 0.09 ^e^	43.37 ± 0.07 ^i^	58.80 ± 0.07 ^j^	17.02 ± 0.11 ^b^	18.74 ± 0.08 ^c^	22.80 ± 0.09 ^d^	28.53 ± 0.11 ^g^	27.02 ± 0.08 ^f^	35.72 ± 0.08 ^h^
LA3	52.27 ± 0.17 ^f^	0.03 ± 0.00 ^a^	0.15 ± 0.01 ^a^	0.20 ± 0.01 ^a^	37.34 ± 0.10 ^de^	26.80 ± 0.13 ^b^	37.30 ± 0.17 ^d^	27.05 ± 0.04 ^bc^	37.75 ± 0.15 ^e^	27.46 ± 0.11 ^c^
AA0	0.09 ± 0.00 ^b^	0.07 ± 0.00 ^ab^	0.28 ± 0.01 ^e^	0.06 ± 0.00 ^a^	0.08 ± 0.00 ^ab^	0.07 ± 0.00 ^ab^	0.15 ± 0.01 ^c^	0.18 ± 0.01 ^d^	0.07 ± 0.00 ^ab^	0.06 ± 0.00 ^a^
GA1	0.06 ± 0.00 ^b^	0.05 ± 0.00 ^a^	0.06 ± 0.00 ^b^	0.20 ± 0.01 ^g^	0.09 ± 0.00 ^d^	0.08 ± 0.00 ^c^	0.08 ± 0.00 ^c^	0.06 ± 0.00 ^b^	0.11 ± 0.00 ^e^	0.13 ± 0.00 ^f^
ΣSFA	10.42 ± 0.02 ^d^	5.93 ± 0.07 ^a^	14.20 ± 0.13 ^h^	15.86 ± 0.06 ^i^	8.75 ± 0.01 ^c^	7.88 ± 0.13 ^b^	11.39 ± 0.02 ^e^	12.21 ± 0.05 ^f^	11.96 ± 0.15 ^f^	12.91 ± 0.06 ^g^
ΣMUFA	22.97 ± 0.11 ^a^	70.46 ± 0.16 ^i^	42.27 ± 0.20 ^g^	25.13 ± 0.12 ^c^	36.81 ± 0.13 ^f^	46.57 ± 0.18 ^h^	28.50 ± 0.10 ^d^	32.20 ± 0.12 ^e^	23.27 ± 0.08 ^a^	23.89 ± 0.08 ^b^
ΣPUFA	66.60 ± 0.08 ^j^	23.60 ± 0.09 ^a^	43.52 ± 0.06 ^b^	59.00 ± 0.06 ^f^	54.36 ± 0.21 ^d^	45.54 ± 0.05 ^c^	60.10 ± 0.08 ^g^	55.58 ± 0.07 ^e^	64.76 ± 0.07 ^i^	63.18 ± 0.03 ^h^
AI	0.12 ± 0.01	0.06 ± 0.01	0.16 ± 0.03	0.19 ± 0.03	0.10 ± 0.01	0.08 ± 0.005	0.13 ± 0.01	0.14 ± 0.01	0.14 ± 0.01	0.15 ± 0.01
TI	0.06 ± 0.005	0.12 ± 0.01	0.32 ± 0.04	0.37 ± 0.04	0.06 ± 0.005	0.07 ± 0.004	0.08 ± 0.003	0.10 ± 0.008	0.08 ± 0.007	0.11 ± 0.009
ALPHA (mg/100 g)	0.96 ± 0.04 ^d^	2.38 ± 0.06 ^g^	0.10 ± 0.01 ^a^	1.09 ± 0.04 ^d^	1.43 ± 0.03 ^e^	1.70 ± 0.04 ^f^	0.67 ± 0.03 ^c^	0.51 ± 0.02 ^b^	1.03 ± 0.06 ^d^	1.04 ± 0.05 ^d^
BETA (mg/100 g)	0.00 ± 0.00 ^a^	3.08 ± 0.06 ^d^	0.00 ± 0.00 ^a^	0.00 ± 0.00 ^a^	0.91 ± 0.04 ^b^	1.53 ± 0.02 ^c^	0.00 ± 0.00 ^a^	0.00 ± 0.00 ^a^	0.00 ± 0.00 ^a^	0.00 ± 0.00 ^a^
GAMMA (mg/100 g)	30.85 ± 0.13 ^d^	37.36 ± 0.11 ^g^	38.98 ± 0.18 ^h^	11.27 ± 0.11 ^a^	31.35 ± 0.09 ^d^	32.42 ± 0.17 ^e^	33.06 ± 0.11 ^e^	34.16 ± 0.13 ^f^	22.18 ± 0.40 ^c^	15.79 ± 0.19 ^b^
IP (h)	7.15	55.17	33.47	30.49	9.23	11.42	8.31	10.53	8.56	10.35

Different letters in the rows of this table indicate statistically significant differences between the means of the samples, determined at a significance level of *p* < 0.05, obtained using Tukey’s HSD test. F—flaxseed oil; A—apricot kernel oil; S—sesame seed oil; BC—black cumin oil; FA73, FA50, FS73, FS50, FBC73, FBC50—oil blends in proportions 70:30 and 50:50; MA0—myristic acid (C14:0); PA0—palmitic acid (C16:0); PA1—palmitoleic acid (C16:1); HE0—heptadecanoic acid (C17:0); HE1—heptadecenoic acid (C17:1); SA0—stearic acid (C18:0); OA1—oleic acid (C18:1); LA2—linoleic acid (C18:2); LA3—linolenic acid (C18:3); AA0—arachidic acid (C20:0); GA1—gondoic acid (C20:1); ΣSFA—sum of saturated fatty acids; ΣMUFA—sum of monounsaturated fatty acids; ΣPUFA—sum of polyunsaturated fatty acids; ALPHA—α tocopherol; BETA—β tocopherol; GAMMA—γ tocopherol; IP (h)—induction period, measured in hours.

**Table 2 foods-14-02000-t002:** The content of phenolic compounds and antioxidant capacity of the oils.

Oil/Blend	Phenolics mg GAE/g Oil	Antioxidant Capacity μmol TE/g Oil
F	1.0	25 ± 1.4
A	0.8	20 ± 1.0
S	1.2	30 ± 1.6
BC	2.5	75 ± 3.5
FA73	0.9	24.7 ± 1.2
FA50	0.9	24.5 ± 1.3
FS73	1.1	28.3 ± 1.4
FS50	1.1	30.5 ± 1.5
FBC73	1.4	44.5 ± 2.2
FBC50	1.8	57.5 ± 2.4

**Table 3 foods-14-02000-t003:** Principal phenolic compounds.

Oil	Principal Phenolics	Amount mg/g Oil
F	Secoisolariciresinol diglucoside	0.29
A	Hydroxycinnamic acids	0.20
	Flavonoids	0.17
S	Sesamin	4.10
	Sesamolin	0.85
BC	Thymoquinone	2.10
	Carvacrol	0.20

## Data Availability

The original contributions presented in this study are included in the article/Appendix A. Further inquiries can be directed to the corresponding author.

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
