# Peer review of "Enhancing Oxidative Stability and Nutritional Quality of Flaxseed Oil Using Apricot, Sesame, and Black Cumin Oil Blends"

_foods, 2025, doi:10.3390/foods14112000_

Round 1

Reviewer 1 Report

Comments and Suggestions for Authors

General Comment: The study addresses a relevant topic in food science by investigating strategies to improve the oxidative stability of flaxseed oil through blending with other functional vegetable oils. The work presents a well-structured experimental approach and employs appropriate methodologies to evaluate physicochemical and antioxidant parameters. However, the manuscript requires improvements in textual organization, clarity of scientific language, more robust justifications for sample selection, and a more critical interpretation of the results. Additionally, grammatical revision and formatting adjustments are needed throughout the text. Some statements lack up-to-date references or are discussed too generically.

Title: Consider rephrasing to: "Enhancing Oxidative Stability and Nutritional Quality of Flaxseed Oil Using Apricot, Sesame, and Black Cumin Oil Blends"

Abstract: Needs to be revised to include numerical results.

Introduction: Reorganize the paragraphs to present: Justification of the nutritional importance of flaxseed oil; Issues associated with its oxidative instability; Functional potential of the complementary oils used; Study objective. Update some of the cited references to include more recent reviews on the topic (2020–2025).

Methodology: Include the number of experimental replicates (n) for each analysis (e.g., in triplicate?). Indicate whether normality tests were performed before applying statistical analyses. Provide full brand names and models of the equipment used.

Results and Discussion: The section presents good data, but the interpretation is superficial in several areas, and correlations are not always critically discussed: Discuss plausible molecular mechanisms that contribute to the improved stability provided by different oils (e.g., presence of tocopherols, sesamin, specific phenolic compounds); Provide a more detailed explanation as to why apricot kernel oil showed better performance, based on its chemical composition; Avoid excessive repetition and generalized statements such as “this is expected” or “clearly shows.”

Conclusion: Include key numerical findings that support the conclusions (e.g., % increase in oxidative stability); Point out the limitations of the study, such as the exclusive use of binary blends or the absence of sensory evaluation.

Reviewer 2 Report

Comments and Suggestions for Authors

This manuscript presents a novel and engaging subject related to sesame seed properties, with a well-structured framework and clear presentation. However, several revisions are necessary:

  1. The abstract should be rewritten to include numerical and quantitative results.
  2. Lines 90-94: More information about the chemical properties of sesame needs to be added to the text. I recommend referring to the publication "Effect of moisture content and temperature on thermal behaviour of sesame seed" to enhance this section with relevant data on sesame's compositional and thermal characteristics.
  3. The novelty of the work is not sufficiently highlighted. Please add a paragraph at the end of the introduction that clearly articulates how this research advances the field and differentiates itself from existing literature.
  4. Section 2.10: The experimental methodology lacks information on replication. Please specify how many times each experimental test was repeated.
  5. The paper would benefit from a more comprehensive comparison of the obtained results with similar research.

Reviewer 3 Report

Comments and Suggestions for Authors

              Comments on the manuscript:

  1. What compound did the authors have in mind when they mentioned compounds, specifically 4-hydroxyenal, which can be formed from lipids and is subject to peroxidation? (line 40)
  2. Please provide examples of omega-3 and omega-6 acids present in linseed oil. (lines 54-55)
  3. Please explain why the lipid content of fish changes during processing. What processing did the authors have in mind, and which lipid content changes? (lines 59-60)
  4. The purpose of the study was specified correctly, but please provide more precise details, i.e. which secondary metabolites were analyzed.
  5. What does "ethanol-based control" mean? (line 182)
  6. In what units were the total phenolic compounds content and antioxidant activity expressed?
  7. What isomers of oleic and linoleic acids did the authors have in mind?
  8. When discussing the results, please use the full names of fatty acids rather than their abbreviations.
  9. The authors wrote that "The oxidative stability of the oil, expressed as induction period (IP) in hours (h), was tested using an Oxitest device. As shown in Figure 1, A oil exhibited the longest induction period (55.17 h), followed by S oil (33.47 h), BC oil (30.49 h) and, as expected, F oil had the shortest induction period (7.15 h)."Where were such IP values ​​included in the manuscript?
  10. Could the authors indicate which linseed oil blend is the most promising considering the parameters tested, i.e. oxidative stability, antioxidant activity, polyphenol content and nutritionally beneficial fatty acid composition? Please comment.

Round 2

Reviewer 1 Report

Comments and Suggestions for Authors

The authors have improved the quality of the study, it can be accepted in its current form.

Reviewer 3 Report

Comments and Suggestions for Authors

I have no additional comments.